biotechnology/ecology

Aquatain, *Anopheles arabiensis*, *Ochlerotatus caspius*, mosquito, AMF, Sudan

**Author for correspondence:**
Rasha Siddig Azrag
e-mail: razrag@hotmail.com

# Comparison of the temporal efficacy of Aquatain surface films for the control of *Anopheles arabiensis* and *Ochlerotatus caspius* larvae from Sudan

Alaa Mahmoud Ali Almalik[1], R. Guy Reeves[2] and Rasha Siddig Azrag[1]

[1]Vector Genetics and Control Laboratory, Department of Zoology, Faculty of Science, University of Khartoum, Sudan
[2]Max Planck Institute for Evolutionary Biology, Plön Germany

AMAA, 0000-0002-5747-5221; RGR, 0000-0001-9454-6175; RSA, 0000-0001-8222-7441

Aquatain mosquito formulation (AMF) is a surfactant that spreads across the surface of water bodies to produce a monomolecular film. This study experimentally evaluates the temporal efficacy of AMF against aquatic stages of *Anopheles arabiensis* and *Ochlerotatus caspius* under laboratory conditions. Using the recommended application dose of 1 ml m$^{-2}$, a large species-specific difference in the median lethal time for L3–L4 larvae was observed. The median lethal time to 50% mortality (LT50) and 90% mortality (LT90) was 1.3 h, 95% CI [1.2, 1.4] and 3.8 h, 95% CI [3.6, 4.0], respectively, for *Oc. caspius*. The corresponding values for *An. arabiensis* were 8.1 h, 95% CI [7.3, 9.0] and 59.6 h, 95% CI [48.5, 76.2]. Based on data from published laboratory studies for a total of seven mosquito species, drawn from four genera, results in the following three groups, [LT50 = 1–2 h, *Culex quinquefasciatus*, *Ochlerotatus caspius*] [LT50 = 8–24, hours, *Anopheles minimus*, *Anopheles arabiensis*, *Anopheles gambiae* s.s.] and [LT50 = 72–143 h, *Anopheles stephensi*, *Aedes aegypti*]. In all experiments, 100% mortality was achieved given sufficient time. The potential relevance of mortality rate estimates, in the context of other studies, on the use of monomolecular films for the control of malaria and arbovirus diseases is discussed.

# 1. Introduction

There is a growing realization that new vector interventions need to be added to the core control tools that target indoor adult mosquitoes. According to the WHO [1], larval source management (LSM) is potentially suitable as a supplement to core interventions for some clearly delineated habitats, particularly in urban areas. Fillinger & Lindsay [2] showed that LSM can be an effective tool in selected eco-epidemiological conditions, as it targets both outdoor and indoor biting/resting mosquitoes equally, which is a critical issue for addressing residual transmission [3–6]. Therefore, it can be an important component in integrated vector management packages [2].

Numerous effective control methods against the larval stages of mosquito vectors of malaria and arboviruses have been developed. The application of many is hindered by economic costs, genetic resistance and environmental concerns. Monomolecular films (MMFs) that consist of non-ionic surfactants were developed as potential alternatives to petroleum-based oils for mosquito control. Their mode of action is through lowering the water surface tension [7,8]. The physical nature of its mode of action makes the selection of genetic resistance less likely than that frequently arising for conventional chemical insecticides. Monomolecular layers/surfactants differ from most other mosquito control agents because of their ability to target multiple stages in the mosquito life cycle. All stages that come in contact with the water surface, e.g. eggs, larvae, pupae, emerging adults and ovipositing females, can be affected by the lowered surface tension caused by such layers [8,9].

Aquatain mosquito formulation (AMF) products are silicon-based liquids that rapidly spread across the surface of bodies of water as a monomolecular layer. AMF is principally composed of the chemical polydimethylsiloxane that is widely used, including as a human food additive (INS no. 900a), a condom lubricant and cosmetics constituent. AMF has been reported to be biodegradable and relatively safe to surveyed non-target invertebrates and vertebrates [9,10]. Applications are recommended to be repeated every four weeks [11] and provide high levels of larval control in a wide range of circumstances [12] including large size slurry rice paddies [13]. As such, the application of AMF provides an increasingly established strategy for larval control including potentially against malaria vectors in Africa [10], as part of integrated vector management programmes.

Numerous dose–response studies have demonstrated the efficacy of AMF against larvae of many malaria vectors including *Anopheles arabiensis*, *An. gambiae* s.s., *An. stephansi* and *An. minimus* [14–18] and major vectors of arboviruses including *Aedes aegpti* [14,18–20], *Ae. Albopictus* [21–24] and *Culex pipens* complex [21–23]. This study seeks to determine the speed of the larvicidal effect of AMF against *An. arabiensis* and *Ochlerotatus caspius* larvae from Sudan under controlled laboratory conditions.

# 2. Materials and methods

With oral permission, mosquito larvae were collected from Kuku Dairy Project at the locality of the East Nile, Khartoum state at latitude 15°39′ N and longitude 32°35′ E. In the laboratory, *Oc. caspius* and *An. arabiensis* late third or fourth instar larvae were sorted out and used in the experiments. Experiments were performed in glass boxes with differing surface areas when filled with 1000 ml of tap water. Aquatain® mosquito formulation (AMF) was added to each glass box according to the manufacturer's recommended concentration of 1 ml m$^{-2}$. AMF was provided by the manufacturer Aquatain Products Pty Ltd., Australia and contains 78% polydimethylsiloxane. The three types of vertical-sided glass boxes used in the experiment are referred to throughout this manuscript as 9 cm, 7 cm and 5 cm (though each also differs in their surface area 115 cm$^2$, 145 cm$^2$ and 200 cm$^2$, respectively). All dimensions below are in cm unless otherwise stated and follow the sequence: width, height and depth [25].

9 cm box: Water depth 8.8 cm, dimensions $16.2 \times 7.1 \times 10.2$, surface area $= 16.2 \times 7.1 = 115$ cm$^2$, 11.5 µl of AMF added.

7 cm box: Water depth 7 cm, dimensions $18 \times 8.1 \times 8.2$, surface area $= 18 \times 8.1 = 145$ cm$^2$, 14.5 µl of AMF added.

5 cm box: Water depth 5.2 cm, dimensions $20 \times 10 \times 6.5$, surface area $= 20 \times 10 = 200$ cm$^2$, 20 µl of AMF added.

AMF was added using a Gilson P20 pipette. Boxes were left overnight before the start of each experiment to stabilize. For each species, three replicates of each box type were performed contemporaneously, along with three control containers of 7 cm depth with no AMF added. Resulting in a total of nine experimental replicates and three control replicates per species.

Fifty *Oc. caspius* or 15 *An. arabiensis* late third or fourth instar larvae were placed in each test glass box and the control boxes. The difference in the larval density between species was dictated by larval availability. The number of dead larvae was recorded hourly for *Oc. caspius* until the 8th hour, by which time 100% mortality had been achieved in all experimental containers (observations at 0, 1, 2, 3, 4, 5, 6, 7, 8 h). For *An. arabiensis*, the number of dead larvae was recorded hourly for the first 12 h then an unsampled period of 60 h and then every 24 h until 100% mortality had been achieved in all experimental containers (observations at 0, 1, 2, 3, 4, 5, 6, 7, 10, 11, 12, [60 h gap], 72, 96, 120 h). Larvae were considered dead if they sank to the bottom of the clear glass boxes and failed to move or float on the surface. No food was added to either tests or controls.

Statistical analysis was performed using JMP 15.1.0 with median lethal time for L3–L4 larvae calculated using a probit approach, and a generalized linear model, with a binomial distribution and a probit link. The full model nests replicate within containers (electronic supplementary material, figure S2). Due to only single larvae dying in any of the six controls, probit mortality estimates were not adjusted using the controls (the single control mortality was *An. arabiensis* replicate R2). Probit *p*-values which can be challenging to represent when they are small are expressed as 'LogWorth' values which is log10(*p*-value).

A literature search was conducted to identify studies reporting median lethal time to 50% mortality (LT50) estimates using 1 ml m$^{-2}$ AMF on L3 and L4 larva, where the surface area of the container was also reported. This resulted in four studies being identified [16,18–20]. In the case of Wang *et al.* [20] and Webb *et al.* [19], it was necessary to reanalyse the data reported in these papers to achieve the stated focus in terms of the larval stage and AMF usage. Because in both studies, graphs were presented without numerical values these were estimated using WebPlotDigitizer v. 4.2 [26] (see electronic supplementary material, S2 file). LT50 estimates were done with the same generalized linear model framework without incorporating container or replicate into the model and not accounting for control mortality. For additional details see electronic supplementary material, table S1 and figure S1.

All data analysed (including reanalysed data) is made available in a single electronic supplementary material, file S2

# 3. Results

A total of 585 larvae (450 *Oc. caspius* and 135 *An. arabiensis*) were exposed to 1 ml m$^{-2}$ of AMF in three types of containers (depth 5 cm, 7 cm and 9 cm) with three replicates each (figure 1). Controls, using the 7 cm container and no AMF, indicated almost no larval background mortality. Only one out of 45 *An. arabiensis* larvae died and 0 out of 150 *Oc. caspius* larvae. Eventually all larvae in experimental containers died, as indicated by their sinking to the bottom and failing to move, or floating inert on the surface. It was also observed that any pupae that developed in experimental treatments failed to moult and also died. The high visibility of these all glass containers facilitated precise measurements (figure 1). All of the 195 pupae, except one, in the control boxes emerged as adults, see controls in figure 2. In the experimental containers both species showed a strong relationship between time and mortality, with 100% mortality being achieved in all containers, figure 2 and 3. However, the rate at which this occurred was very different between species, with all *Oc. caspius* larvae dead within 8 h, but all *An. arabiensis* larvae took more than 3 days. Using a probit analysis, LT50 and LT90 estimates were made for each replicate separately (*n* = 50 *Oc. caspius* or *n* = 15 *An. arabiensis*), mean for each container (*n* = 150 *Oc. caspius* or *n* = 45 *An. arabiensis*) and for both species (*n* = 450 *Oc. caspius* or *n* = 135 *An. arabiensis*). All these estimates are shown in figure 3 and detailed in electronic supplementary material, table S2. With the exception of the *An. arabiensis* replicate R3 in the 5 cm container, there is a high degree of internal consistency between the replicate estimates and also with the mean estimates (figure 3). The median lethal time to 50% mortality (LT50) and 90% mortality (LT90) was 1.3 h, 95% CI [1.2, 1.4] and 3.8 h, 95% CI [3.6, 4.0], respectively, for *Oc. caspius*. The corresponding values for *An. arabiensis* were 8.1 h, 95% CI [7.3, 9.0] and 59.6 h, 95% CI [48.5, 76.2].

It should be noted that the LT90 estimate for *An. arabiensis* has by far the broadest confidence intervals (CI) and this is likely to reflect that the 59.6 h estimate falls within the 60 h window without any observations. To test for statistical significance and to quantify the relative effect of the container and replicates within each of the two species, the probit nested model described in the Materials and methods section and electronic supplementary material, figure S2 was conducted. The results in table 1 confirm the time as the major explanatory variable of mortality for both species LogWorth of *Oc. caspius* = 206.4 *An. arabiensis* = 139.6 both of which are highly significant. The LogWorth impact of the container type

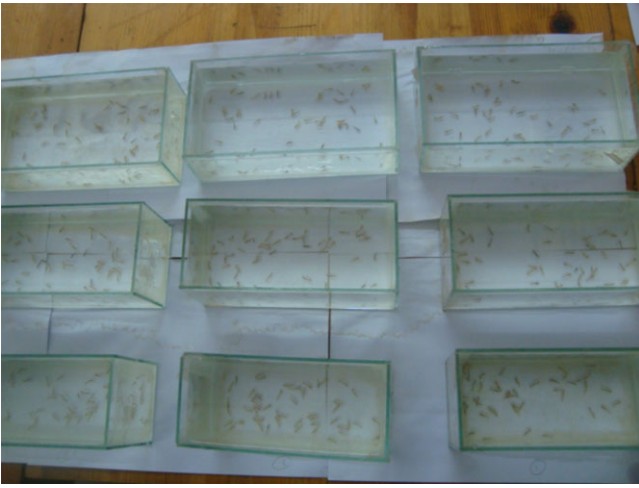

**Figure 1.** Containers and larvae used in experiments. Top view of the nine experimental containers with *Ochlerotatus caspius* larvae.

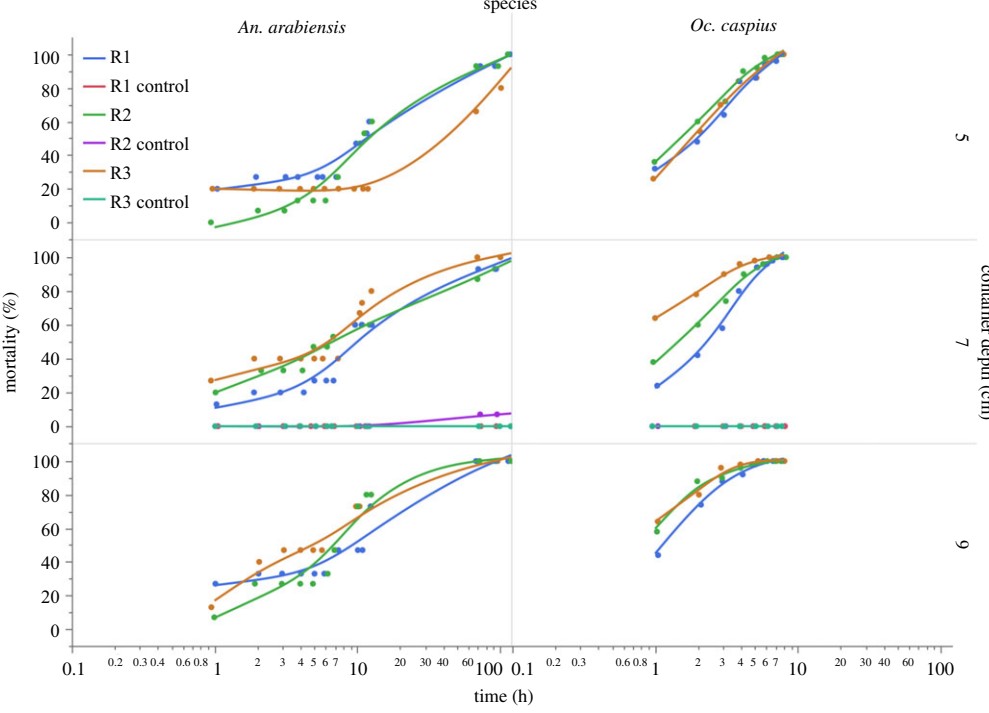

**Figure 2.** Percentage mortality against experimental duration for *Anopheles arabiensis* and *Ochlerotatus caspius* L3–L4 larvae in the three containers used in this study. In both species and across all three types of containers (5, 7 and 9 cm depth) and all 18 experimental replicates (AMF used at 1 ml m$^{-2}$), 100% mortality was achieved. However, the rate of mortality was substantially different between the two species; note the log10 scale of the x-axes. All replicates started at time 0 with 0% mortality, though this time point is not shown. Three replicated contemporaneous controls (no AMF) were performed for both species using the 7 cm depth container; these are shown on the graphs but have limited visibility as in only one of the six replicates was any mortality observed (*An. arabiensis* R2 control). Replicate regression lines are consistent but arbitrary and only to indicate the trends in each replicate. Data points have been jittered to increase the visibility of measurements taken at the same time point.

used is greater than 10-fold smaller *Oc. caspius* = 22.0 *An. arabiensis* = 11.7, though again highly significant. The impact of replicates (nested within containers) is again significant, but has a relatively small effect size.

Comparisons between LT50 estimates from the literature combined with the results of this study showed a noticeable inter- and intra-specific variation between different mosquito genera or species (figure 4). *Ochlerotatus* and *Culex* genera showed the lowest LT50 estimates followed by the *Anopheleles* genus while the *Aedes* genus showed the highest LT50 estimates.

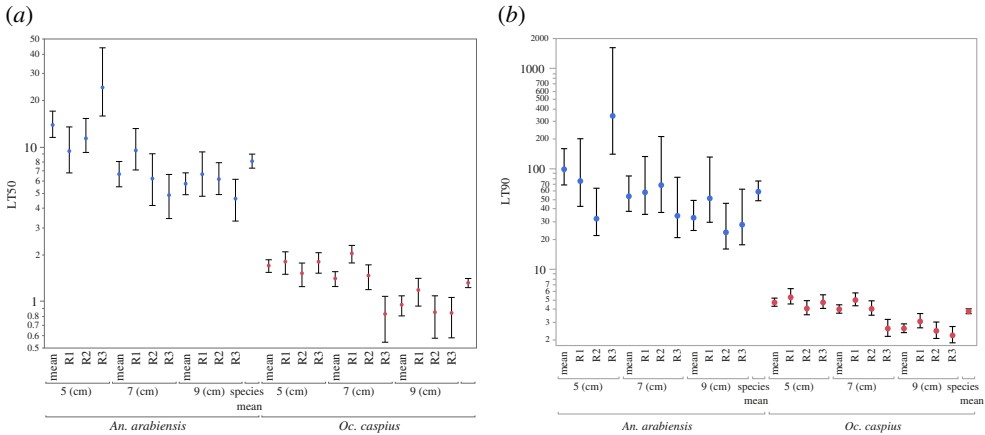

**Figure 3.** Estimates of LT50 and LT90 of *Anopheles arabiensis* and *Ochlerotatus caspius* L3–L4 larvae. Probit estimates are provided for all replicates (*n* = 3, R1, R2, R3) for each of the three container depths used (cm). Corresponding means for containers (*n* = 3) and overall species means (*n* = 9) are shown. *Anopheles arabiensis*, the least surfactant sensitive of the two species, is shown in blue, and *Ochlerotatus caspius* is shown in red. Bars represent the 95% probit confidence estimates. Full data plotted given in electronic supplementary material, table S2.

**Table 1.** Species-specific significance and effect size of generalized linear probit model. Time, as a continuous variable, is the major explanatory variable, with relatively modest (though significant) impacts of container and replicate (the latter as a variable nested within the former). *p*-value and ChiSquare are for likelihood ratio test, d.f. are the degrees of freedom used. Note that the *p*-values in this table indicate whether the variable in question is likely to be an explanatory one, and do not directly relate to the confidence intervals for the LT estimates shown in figure 3.

| species | variable | LogWorth[a] | d.f. | L-R ChiSquare[b] | *p*-value |
|---|---|---|---|---|---|
| *An. arabiensis* | log(time_hours) | 139.6 | 1 | 636.0 | $2.5 \times 10^{-140}$ |
| | container | 11.7 | 2 | 53.9 | $2.0 \times 10^{-12}$ |
| | replicate[container] | 3.4 | 6 | 24.7 | $3.9 \times 10^{-04}$ |
| *Oc. caspius* | log(time_hours) | 206.4 | 1 | 943.0 | $4.5 \times 10^{-207}$ |
| | container | 22.0 | 2 | 101.1 | $1.1 \times 10^{-22}$ |
| | replicate[container] | 12.0 | 6 | 69.8 | $4.6 \times 10^{-13}$ |

[a]LogWorth = log10(*p*-value).
[b]L-R ChiSquare = the value of the likelihood ratio ChiSquare statistic for a test of the corresponding effect.

## 4. Discussion

The need for new and innovative methods for mosquito control is currently increasing due to high insecticide resistance and behavioural changes. AMF's mode of action as a surfactant, lowering of water surface tension, may limit the capacity for insecticide resistance to evolve [11,12]. As in prior studies, it was confirmed that surfactants such as AMF, using recommended application doses and with sufficient time, achieve 100% mortality. This was done for the first time for *Oc. caspius* in addition to *An. arabiensis* from Sudan using their late larval stages, which have repeatedly been shown to be the more resistant stages to surfactants in a number of species [16,24]. The striking difference in both the LT50 and LT90 estimates between species (*Oc. caspius* and *An. arabiensis*) under constant laboratory conditions is consistent with previous reports also demonstrating large differences between pairs of species [13,14,18,19,21,22]. We cannot, however, discount the possibility that at least some of this difference could be due to the single experimental variable that was not held constant for both species, namely larval density. However, we consider this as an unlikely major explanatory factor as both densities are well below what might be considered overcrowded for mosquito larvae in terms of water volume or surface area (figure 1). Larval density in terms of water volume for *Oc. caspius* was one larva per 20 ml, and for *An. Arabiensis*, it was 1 larvae per 66 ml. Both are well below the maximal density of WHO guidelines of one larva per 4 ml, recommended to facilitate comparisons between experiments assessing larvicides [27, p. 10]. The possibility that larval 'undercrowding' of *An. arabiensis* is a factor in the large

**Figure 4.** Literature LT50 estimates for AMF. Estimates from all five available laboratory studies where Aquatain is used at its recommended $1\,ml\,m^{-2}$. While there are differences in the experimental procedures between studies (see electronic supplementary material, table S1), it is probably still meaningful to discern the following observations. (i) In the four studies where more than one species was examined all demonstrated substantial variation exists between species under the same conditions (greater than threefold), (ii) For the single species where more than one independent estimate is available (*A. aegypti*), all three independent estimates are broadly similar. (iii) For the single genus where multiple estimates for more than one species is available (*Anopheles*), the estimates suggest that a broad range of species-specific sensitivities can be represented within a genus (greater than eightfold).

observed difference between the species, again cannot be discounted, but as far as we are aware it has never been reported as a factor in other experiments or testing guidelines.

The relationship between mortality rate and container depth is negative in both species, with shallower containers having longer LT estimates (see LT50 and LT90 trends in figures 2 and 3). However, given that only three container types were used, it is not possible to identify if it is the container depth, the perimeter length or surface area that is the causative factor. It is potentially notable that a negative correlation observed between mortality rate and container depth is consistent with the idea that the deeper the container the higher the energetic cost to larvae of not being able to float at the surface breathing through their syphon [28]. As, larvae in shallower containers have a shorter vertical distance to travel from their alternative resting position on the bottom of the container than those in deeper containers (figure 1). It is perhaps rather surprising that such a statistically significant relationship (table 1) can be observed between containers that differ by a maximum of 5 cm depth. Consequently, this hypothesis should be treated with considerable caution until additional experimental tests are performed that hold surface area constant and vary depth over a very much wider range. In summary, while based on their mode of action it might be assumed that the temporal efficacy of surfactants should be independent of water depth. If, however, a relationship between water body depth and increased sensitivity to AMF were confirmed, this could have important positive consequences for understanding the effectiveness of field use of surfactants [7,11,14,15,21,24,27].

Placing the LT50 estimates in the context of other studies of late larval stages where $1\,ml\,m^{-2}$ of AMF was applied, provides some tentative general insights (figure 4). First, for the species (*Aedes aegypti*) where multiple estimates have been independently generated, they are rather consistent. This implies that the inevitable differences in experimental conditions between these three studies do not have a major effect (electronic supplementary material, table S1). Second, that the temporal sensitivity, as indicated by LT50 estimates, can vary substantially within genera This is the case for the genus *Anopheles* with a greater than eightfold range of LT50 values between species. Furthermore, the close taxonomic relationship between the *Aedes* and *Ochlerotatus* genera (the former was until recently a sub-genus within the latter) indicates that there exists a substantial approximately 100-fold variation within the Aedini

tribe, of which they are both members. This implies that it may be problematic to extrapolate LT50 estimates for species within the same genus. Currently, there are no ecological or biological hypotheses of what may underlie these striking differences between often closely related species. Though it must be kept in mind that it is conceivable, despite the similarity of the estimates between studies observed for *Aedes aegypti*, that other experimental conditions that differed between studies (electronic supplementary material, table S1) may play a role in establishing the variation presented in figure 4. With respect to this study, it is potentially notable that it is unique (electronic supplementary material, table S1) in not providing food to larvae. While we cannot discount this may have had an impact on LT50 estimates we would note that WHO guidelines indicate that food is only necessary for 'long' experiments [29, p. 10] and the LT50 estimates in this study are both under 9 h. Furthermore, the near-zero mortality observed in controls suggests that starvation was not a major issue for either LT50 or LT90 estimates.

In practical terms, it is unclear what if any significance laboratory-based estimates of temporal sensitivity have for field applications, where AMF is generally applied for weeks at a time (vastly exceeding the LT50 or LT90 for all aquatic stages of mosquito species surveyed to date). In this context, it is relevant that the most resistant species examined in figure 4 is *Aedes aegypti*, where field trials have demonstrated the effectiveness of AMF as a control measure against this species (e.g. [14,15]). However, given that the raw material of evolution is variation, it is interesting to speculate on what selective forces may have shaped the wide range of phenotypic values indicated in figure 4. Potentially, the microlayer at the air–water boundary in the various ecological niches that the mosquito species occupy may be more complicated than generally appreciated and that this could indicate that increased tolerance to surfactant films may evolve, potentially due to exposure to degrading or older monomolecular films. However, other non-adaptive explanations for the variation in figure 4 are also possible [30], and these would not imply a significant capacity for increased surfactant tolerance to evolve within species.

Sudan has an increasing risk of mosquito-borne diseases including malaria [31] and arboviral diseases [32]. Based on our results, we support expanding laboratory and field evaluations of AMF as a suitable tool for inclusion in integrated vector management packages against mosquito vectors. This may be especially valuable in urban settings in Khartoum state and Khartoum city which has a long history of larval control interventions [33], and which is threatened by insecticide resistance to Temephos [34].

# 5. Conclusion

Laboratory dose–response assays demonstrated the efficacy of AMF against *Oc. caspius* and *Anopheles arabiensis* from Sudan as a control measure against both species. Our findings highlighted the noticeable interspecific and intraspecific variation between different mosquito genera/species based on LT50 estimates and the negative relationship between mortality rate and water depth for both species. Further studies are needed for understanding both the temporal sensitivity of AMF between and within different mosquito genera and the temporal efficacy of AMF.

Data accessibility. The dataset supporting this article has been uploaded as part of the electronic supplementary material file S2.

Authors' contributions. Substantial contributions to conception and design, or acquisition of data, or analysis and interpretation of data: A.M.A.A., R.G.R. and R.S.A. Drafting the article or revising it critically for important intellectual content: A.M.A.A., R.G.R. and R.S.A. Final approval of the version to be published: A.M.A.A., R.G.R. and R.S.A. Agreement to be accountable for all aspects of the work in ensuring that questions related to the accuracy or integrity of any part of the work are appropriately investigated and resolved: A.M.A.A., R.G.R. and R.S.A.

Competing interests. We have no competing interests.

Funding. R.G.R. is supported by funds from the Max Planck Society.

Acknowledgements. We would like to thank two anonymous reviewers who significantly improved the manuscript. We gratefully acknowledge the donation of a 500 ml bottle of AMF donated by Aquatain Products Pty Ltd., Australia.

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
