## [Peer Review File · Royal Society Open Science]

Review History

RSOS-200980.R0 (Original submission)

Review form: Reviewer 1

Is the manuscript scientifically sound in its present form?

No

Are the interpretations and conclusions justified by the results?

Yes

Is the language acceptable?

Yes

Do you have any ethical concerns with this paper?

No

Have you any concerns about statistical analyses in this paper?

No

Recommendation?

Accept with minor revision (please list in comments)

Comments to the Author(s)

In this study the authors compare the effect of aquatrain surface films on time to mortality between two closely related mosquito species: *Anopheles arabiensis* and *Ochlerotatus caspius*. Across different container sizes they find the consistent result that *An arabiensis* takes about 15x longer to kill 90% of the larvae than *Och caspius*. This study brings relevant data on the lethal times of these surface films and I support the publication of this manuscript. However, I have a few concerns regarding the experimental setup which limits the ability to draw certain conclusions.

Major comments:

- *An arabiensis* and *Och caspius* were exposed to aquatrain at different larval densities, making this experimental design not suitable to do direct comparisons between species. While unlikely, one cannot exclude the possibility that these results are due to larval density instead of species. This issue needs to be addressed in the discussion.
- Related to the above comment, it would also be helpful to comment on larval densities of other studies and add to table S2.
- No food was given whereas all other studies studying surface films did give food (table S2), please provide rationale for not providing food and how this could have impacted the results.
- Mortality was observed hourly for the first 12 hours and every 24 hours thereafter. This will have impacted accuracy of LT50 and LT90 estimates for *An arabiensis*. However, looking at figure 2, I don't see observations being made between 12 hours and approximately 65 hours. I presume the x-axis is incorrect? Regardless, there is a large gap between the two measurements in which mortality could have rapidly increased leading to an overestimation of time to mortality in *An. arabiensis*. The lack of observation point during increases in mortality is a major limitation to the ability to accurately predict lethal times, which should be discussed.
- There are quite a few grammatical mistakes and typo's in the manuscript. I address some below but note this is not an exhaustive list.

Minor comments:

- Page 2, line 27: replace 'does' with 'dose', add hyphen "species-specific"
- Page 3, line 15: confusing wording. Suggestion "are referred throughout this manuscript as 9cm, 7cm and 5cm"
- line 20: add which is width, height and depth
- line 44-46: confusing wording in methods section, reword to reflect this to be a literature search to identify studies within certain parameter space
- bottom page: "that was not surfactant induced" confusing wording. Suggestion "indicating low background mortality"
- page 4, line 5: [...] pupae that developed" add "in experimental treatments"
- line 14: remove duplicate (n=150 *Oc. caspius* ...)
- line 25: remove "as expected"
- line 25: what is "logWorth"? Explain in methods
- line 30: replace significate with 'significant'
- line 32-50: move to discussion
- page 5, top: 'were' = 'where'
- page 5, line 2: period at end of sentence
- line 11: hyphen in 'laboratory-based'
- line 24: I agree that lowering the water surface tension may limit the capacity for resistance to evolve. However, you also demonstrate that there exist a wide biological variation among species

and variation is the driving force behind evolution. Possibly longer time to mortality could be selected for if larvae are exposed to degrading, older, films?

- Table 1: explain which test was performed and definition of LogWorth and L-R ChiSquare

- Figure 1: Annotate pictures. Picture A would benefit from adding the names of the different containers. I'm not sure what the benefit of pictures B and C are (and not clear what I'm looking at in B).

- Figure 3A: Time should also be plotted on log-scale for consistency and ease of comparisons. Drawing means in different color or thicker lines helps for quick comparisons.

Review form: Reviewer 2

Is the manuscript scientifically sound in its present form?

No

Are the interpretations and conclusions justified by the results?

Yes

Is the language acceptable?

Yes

Do you have any ethical concerns with this paper?

Yes

Have you any concerns about statistical analyses in this paper?

No

Recommendation?

Major revision is needed (please make suggestions in comments)

Comments to the Author(s)

This article presents an interesting research area in the use of monomolecular surface films for larviciding. However, the manuscript requires revisions. The introduction section of the manuscripts will require additional input to review in greater detail related published works. Also consider, discussing your results in much greater depth in the discussion section to allow interpretation of your research findings. Check on your references and ensure that appropriate referencing is done.

Decision letter (RSOS-200980.R0)

Dear Dr Reeves,

The editors assigned to your paper ("Comparison of the temporal efficacy of Aquatain surface films for the control of *Anopheles arabiensis* and *Ochlerotatus caspius* larvae from Sudan") have now received comments from reviewers. We would like you to revise your paper in accordance

with the referee and Associate Editor suggestions which can be found below (not including confidential reports to the Editor). Please note this decision does not guarantee eventual acceptance.

Please submit a copy of your revised paper before 05-Aug-2020. Please note that the revision deadline will expire at 00.00am on this date. If we do not hear from you within this time then it will be assumed that the paper has been withdrawn. In exceptional circumstances, extensions may be possible if agreed with the Editorial Office in advance. We do not allow multiple rounds of revision so we urge you to make every effort to fully address all of the comments at this stage. If deemed necessary by the Editors, your manuscript will be sent back to one or more of the original reviewers for assessment. If the original reviewers are not available, we may invite new reviewers.

- Data accessibility

<http://datadryad.org/submit?journalID=RSOS&manu=RSOS-200980>

- Competing interests

- Authors' contributions

- Acknowledgements

- Funding statement

on behalf of Dr Krijn Paaijmans (Associate Editor) and Pete Smith (Subject Editor)
openscience@royalsociety.org

Associate Editor's comments (Dr Krijn Paaijmans):

Associate Editor: 1

Comments to the Author:

Dear authors,

Both reviewers raised some concerns that need to be addressed:

- 1) The fact that different larval densities have been used for the two species, so the question is whether your observation is a result of the different species or simply an effect of density
- 2) Food conditions: no food was provided, but how will this affect the results?
- 3) Additional review of the published literature is required in the introduction and discussion sections. A better interpretation of your results and how comparable these are to previous studies is warranted.
- 4) Did you obtain ethical approval (and written or verbal permission) to collect larvae from the Kuku Dairy Project?

Comments to Author:

Reviewers' Comments to Author:

Reviewer: 1

Comments to the Author(s)

In this study the authors compare the effect of aquatrain surface films on time to mortality between two closely related mosquito species: *Anopheles arabiensis* and *Ochlerotatus caspius*. Across different container sizes they find the consistent result that *An arabiensis* takes about 15x longer to kill 90% of the larvae than *Och caspius*. This study brings relevant data on the lethal times of these surface films and I support the publication of this manuscript. However, I have a few concerns regarding the experimental setup which limits the ability to draw certain conclusions.

Major comments:

- *An arabiensis* and *Och caspius* were exposed to aquatrain at different larval densities, making this experimental design not suitable to do direct comparisons between species. While unlikely, one cannot exclude the possibility that these results are due to larval density instead of species. This issue needs to be addressed in the discussion.
- Related to the above comment, it would also be helpful to comment on larval densities of other studies and add to table S2.
- No food was given whereas all other studies studying surface films did give food (table S2), please provide rationale for not providing food and how this could have impacted the results.
- Mortality was observed hourly for the first 12 hours and every 24 hours thereafter. This will have impacted accuracy of LT50 and LT90 estimates for *An arabiensis*. However, looking at figure 2, I don't see observations being made between 12 hours and approximately 65 hours. I presume the x-axis is incorrect? Regardless, there is a large gap between the two measurements in which mortality could have rapidly increased leading to an overestimation of time to mortality in *An. arabiensis*. The lack of observation point during increases in mortality is a major limitation to the ability to accurately predict lethal times, which should be discussed.
- There are quite a few grammatical mistakes and typo's in the manuscript. I address some below but note this is not an exhaustive list.

Minor comments:

- Page 2, line 27: replace 'does' with 'dose', add hyphen "species-specific"
- Page 3, line 15: confusing wording. Suggestion "are referred throughout this manuscript as 9cm, 7cm and 5cm"
- line 20: add which is width, height and depth
- line 44-46: confusing wording in methods section, reword to reflect this to be a literature search to identify studies within certain parameter space
- bottom page: "that was not surfactant induced" confusing wording. Suggestion "indicating low background mortality"
- page 4, line 5: [...] pupae that developed" add "in experimental treatments"
- line 14: remove duplicate (n=150 *Oc. caspius* ...)
- line 25: remove "as expected"
- line 25: what is "logWorth"? Explain in methods
- line 30: replace significate with 'significant'
- line 32-50: move to discussion
- page 5, top: 'were' = 'where'
- page 5, line 2: period at end of sentence
- line 11: hyphen in 'laboratory-based'
- line 24: I agree that lowering the water surface tension may limit the capacity for resistance to evolve. However, you also demonstrate that there exist a wide biological variation among species and variation is the driving force behind evolution. Possibly longer time to mortality could be selected for if larvae are exposed to degrading, older, films?

- Table 1: explain which test was performed and definition of LogWorth and L-R ChiSquare
- Figure 1: Annotate pictures. Picture A would benefit from adding the names of the different containers. I'm not sure what the benefit of pictures B and C are (and not clear what I'm looking at in B).
- Figure 3A: Time should also be plotted on log-scale for consistency and ease of comparisons. Drawing means in different color or thicker lines helps for quick comparisons.

Reviewer: 2

Comments to the Author(s)

This article presents an interesting research area in the use of monomolecular surface films for larviciding. However, the manuscript requires revisions. The introduction section of the manuscripts will require additional input to review in greater detail related published works. Also consider, discussing your results in much greater depth in the discussion section to allow interpretation of your research findings. Check on your references and ensure that appropriate referencing is done.

Author's Response to Decision Letter for (RSOS-200980.R0)

See Appendix A.

RSOS-200980.R1 (Revision)

Review form: Reviewer 1

Is the manuscript scientifically sound in its present form?

Yes

Are the interpretations and conclusions justified by the results?

Yes

Is the language acceptable?

Yes

Do you have any ethical concerns with this paper?

No

Have you any concerns about statistical analyses in this paper?

Yes

Recommendation?

Accept with minor revision (please list in comments)

Comments to the Author(s)

I would like to applaud the authors for incorporating the suggestions made on their manuscript. I believe the discussion is more useful now for the reader to interpret the data and the conclusions that are drawn are better incorporating the limitations of the experimental design.

I only have a few minor comments:

- The manuscript has greatly improved in its readability in terms of typos and grammatical mistakes, I would only suggest to focus on punctuation (eg comma placements).

- I'm not too familiar with the type of analysis that has been done. However, looking at figures 2 and 3 and the 95% confidence intervals presented in the supplementary materials, I'm confused by the ultra-low p-values reported. For instance, there is quite some overlap between replicate 95CI, why these extremely low p-values? It would be useful to have the correctness of the statistical approach verified, particularly with these highly skewed log-scale data. However, I also note that these analyses will not change the conclusion.

Review form: Reviewer 2

Is the manuscript scientifically sound in its present form?

Yes

Are the interpretations and conclusions justified by the results?

Yes

Is the language acceptable?

Yes

Do you have any ethical concerns with this paper?

No

Have you any concerns about statistical analyses in this paper?

No

Recommendation?

Accept with minor revision (please list in comments)

Comments to the Author(s)

There is a manuscript that provides important data on the use of surface films for mosquito control. The manuscript details the important contribution that surface films can play in the management of insecticide resistance, one of the major challenges in malaria control. Field studies would provide the required information on the operational use of the surface films for mosquito control.

Decision letter (RSOS-200980.R1)

Dear Dr Reeves

On behalf of the Editors, we are pleased to inform you that your Manuscript RSOS-200980.R1 "Comparison of the temporal efficacy of Aquatain surface films for the control of *Anopheles arabiensis* and *Ochlerotatus caspius* larvae from Sudan" has been accepted for publication in Royal Society Open Science subject to minor revision in accordance with the referees' reports. Please find the referees' comments along with any feedback from the Editors below my signature.

Please submit your revised manuscript and required files (see below) no later than 7 days from today's (ie 21-Oct-2020) date. Note: the ScholarOne system will 'lock' if submission of the revision is attempted 7 or more days after the deadline. If you do not think you will be able to meet this deadline please contact the editorial office immediately.

on behalf of Dr Krijn Paaijmans (Associate Editor) and Pete Smith (Subject Editor)
openscience@royalsociety.org

Associate Editor Comments to Author (Dr Krijn Paaijmans):

Reviewer 2 recommends that the paper is seen by a statistician, as the 95% confidence intervals in the figures do not rhyme with the ultra-low p-values reported. I suggest the authors check the 95% confidence intervals (and the ways they were calculated) carefully before submission of the final version.

Reviewer comments to Author:

Reviewer: 1

Comments to the Author(s)

I would like to applaud the authors for incorporating the suggestions made on their manuscript. I believe the discussion is more useful now for the reader to interpret the data and the conclusions that are drawn are better incorporating the limitations of the experimental design.

I only have a few minor comments:

- The manuscript has greatly improved in its readability in terms of typos and grammatical mistakes, I would only suggest to focus on punctuation (eg comma placements).

- I'm not too familiar with the type of analysis that has been done. However, looking at figures 2 and 3 and the 95% confidence intervals presented in the supplementary materials, I'm confused by the ultra-low p-values reported. For instance, there is quite some overlap between replicate 95CI, why these extremely low p-values? It would be useful to have the correctness of the statistical approach verified, particularly with these highly skewed log-scale data. However, I also note that these analyses will not change the conclusion.

Reviewer: 2

Comments to the Author(s)

There is a manuscript that provides important data on the use of surface films for mosquito control. The manuscript details the important contribution that surface films can play in the management of insecticide resistance, one of the major challenges in malaria control. Field studies would provide the required information on the operational use of the surface films for mosquito control.

===PREPARING YOUR MANUSCRIPT===

===PREPARING YOUR REVISION IN SCHOLARONE===

Author's Response to Decision Letter for (RSOS-200980.R1)

See Appendix B.

Decision letter (RSOS-200980.R2)

Dear Dr Reeves,

It is a pleasure to accept your manuscript entitled "Comparison of the temporal efficacy of Aquatrain surface films for the control of *Anopheles arabiensis* and *Ochlerotatus caspius* larvae from Sudan" in its current form for publication in Royal Society Open Science. The comments of the reviewer(s) who reviewed your manuscript are included at the foot of this letter.

on behalf of Dr Krijn Paaijmans (Associate Editor) and Pete Smith (Subject Editor)
openscience@royalsociety.org

Associate Editor Comments to Author (Dr Krijn Paaijmans):
Comments to the Author:
Thank you for your work in revising your paper.

Appendix A

Dear editor and Reviewers

We would like to thank the reviewers for the time they took with their enormously constructive comments, we think in addressing them we have considerably improved the manuscript.

We have endeavored to remove all English mistakes, but should this not prove to be the case we would be willing to pay for a professional proof reader.

Thanks

Rasha and Guy

All reviewer and editorial comments are reproduced in full in **black**, with responses in blue

We would like to

1) The fact that different larval densities have been used for the two species, so the question is whether your observation is a result of the different species or simply an effect of density.

Done see Reviewer 1 detailed reply

2) Food conditions: no food was provided, but how will this affect the results?

Done see Reviewer 1 detailed reply

3) Additional review of the published literature is required in the introduction and discussion sections. A better interpretation of your results and how comparable these are to previous studies is warranted.

We have attempted to do this in both the introduction and discussion sections

4) Did you obtain ethical approval (and written or verbal permission) to collect larvae from the Kuku Dairy Project?

we obtained oral permission to collect larvae from KuKu Dairy Project

Comments to Author:

Reviewers' Comments to Author:

Reviewer: 1

Comments to the Author(s)

In this study the authors compare the effect of aquatain surface films on time to mortality between two closely related mosquito species: *Anopheles arabiensis* and *Ochlerotatus caspius*. Across different container sizes they find the consistent result that *An arabiensis* takes about 15x longer to kill 90% of the

larvae than *Och caspius*. This study brings relevant data on the lethal times of these surface films and I support the publication of this manuscript. However, I have a few concerns regarding the experimental setup which limits the ability to draw certain conclusions.

Major comments:

- *An arabiensis* and *Och caspius* were exposed to aquatain at different larval densities, making this experimental design not suitable to do direct comparisons between species. While unlikely, one cannot exclude the possibility that these results are due to larval density instead of species. This issue needs to be addressed in the discussion.

DONE We have the following to the discussion

We cannot however discount the possibility that at least some of this difference could be due to the single experimental variable that was not held constant for both species, namely larval density. However, we consider this as an unlikely major explanatory factor as both densities are well below what might be considered overcrowded for mosquito larvae in terms of water volume or surface area (figure 1). Larval density in terms of water volume for *Oc. caspius* was 1 larvae per 20ml and for *An. arabiensis* it was 1 larvae per 66ml. Both are well below the maximal density of WHO guidelines of 1 larvae per 4ml, recommended to facilitate comparisons between experiments assessing larvicides (page 10, [28]). The possibility that larval “undercrowding” of *An. arabiensis* is a factor in the large observed difference between the species, again cannot be discounted, but as far as we are aware has never been reported as a factor in other experiments or testing guidelines.

- Related to the above comment, it would also be helpful to comment on larval densities of other studies and add to table S2.

DONE “volume of water per larvae (ml)” was added as column

This would enable interested readers to easily generate the following graph, which we speculate is the basis of this request.

This graph shows that given the current data available there is no general trend (however, given the distribution of the data it is not possible to make any confident statements).

Only a single value (3ml per larvae) exceeds the WHO density recommendations.

The red data points are those studies where no food was provided (see next point)

- No food was given whereas all other studies studying surface films did give food (table S2), please provide rationale for not providing food and how this could have impacted the results.

Done the following text was added to the discussion. Note that the two data points in the above graph

Though it must be kept in mind that it is conceivable, despite the similarity of the estimates between studies observed for *Aedes aegypti*, that other experimental conditions that differed between studies (table S1) may play a role in establishing the variation presented in figure 4. With respects to this study it is potentially notable that it is unique (table S1) in not providing food to larvae. While we cannot discount this may have had an impact on LT50 estimates we would note that WHO guidelines indicate that food is only necessary for “long” experiments (page 10, [29]) and the LT50 estimates in this study are both under 9 hours. Furthermore, the near zero mortality observed in controls suggests that starvation was not a major issue for either LT50 or LT90 estimates.

- Mortality was observed hourly for the first 12 hours and every 24 hours thereafter. This will have impacted accuracy of LT50 and LT90 estimates for *An. arabiensis*. However, looking at figure 2, I don't see observations being made between 12 hours and approximately 65 hours. I presume the x-axis is incorrect? Regardless, there is a large gap between the two measurements in which mortality could have rapidly increased leading to an overestimation of time to mortality in *An. arabiensis*. The lack of observation point during increases in mortality is a major limitation to the ability to accurately predict lethal times, which should be discussed.

For *An. arabiensis*

For *An. arabiensis* the number of dead larvae was recorded hourly for the first 12 hours then an un-sampled period of 60 hours and then every 24 hours until 100% mortality had been achieved in all experimental containers (observations at 0, 1, 2, 3, 4, 5, 6, 7, 10, 11, 12, [60 hour gap], 72, 96, 120 hours).

LT50 8.1 hours, 95% CI [7.3, 9.0]

LT90 59.6 hours, 95% CI [48.5, 76.2].

The reviewer is correct there is a gap of 60 hours with no observations. This has now been made very clear in the methods section. The graphs were correct (as was the data file), it was the information in the methods sections was not and we apologies for that.

We consider the fact that we calculate and provide CI intervals for the Probit estimates provides a sound basis to confirm the reviewers insight, at least for *An. arabiensis* LT90 and confirm the basis of the LT50 estimates.

In the results section we have added

It should be noted that the LT90 estimate for An. arabiensis has proportionally by far the broadest confidence intervals (CI) and this is likely to reflect that the 59.6 hour estimate falls within a 60 hour window without any observations.

However if the reviewers prefer we could remove all reference of LT90 from the entire manuscript, though we believe highlighting the issue for the readers and retaining them is the better of the two options.

for *Oc. caspius*

The number of dead larvae was recorded hourly for *Oc. caspius* until the 8th hour, by which time 100% mortality had been achieved in all experimental containers (observations at 0, 1, 2, 3, 4, 5, 6, 7, 8 hours)

LT50 1.3 hours, 95% CI [1.2, 1.4]

LT90 3.8 hours, 95% CI [3.6, 4.0]

- There are quite a few grammatical mistakes and typo's in the manuscript. I address some below but note this is not an exhaustive list.

Minor comments

- Page 2, line 27: replace 'does' with 'dose', add hyphen "species-specific"
done

- Page 3, line 15: confusing wording. Suggestion "are referred throughout this manuscript as 9cm, 7cm and 5cm"

done

- line 20: add which is width, height and depth

done

- line 44-46: confusing wording in methods section, reword to reflect this to be a literature search to identify studies within certain parameter space

done

- bottom page: "that was not surfactant induced" confusing wording. Suggestion "indicating low background mortality"

done

- page 4, line 5: [...] pupae that developed" add "in experimental treatments

done

- line 14: remove duplicate (n=150 *Oc. caspius* ...)

Do not understand ?

- line 25: remove "as expected"

done

- line 25: what is "logWorth"? Explain in methods

done "P-values which can be challenging to represent when they are small, are expressed as "Logworth" values which is $\log_{10}(\text{p-value})$."

- line 30: replace significate with 'significant'

done

- line 32-50: move to discussion

done

- page 5, top: 'were' = 'where'

done

- page 5, line 2: period at end of sentence

- line 11: hyphen in 'laboratory-based'

done

- line 24: I agree that lowering the water surface tension may limit the capacity for resistance to evolve. However, you also demonstrate that there exist a wide biological variation among species and variation is the driving force behind evolution. Possibly longer time to mortality could be selected for if larvae are exposed to degrading, older, films?

Done with pleasure, done at the end of discussion

- Table 1: explain which test was performed and definition of LogWorth and L-R ChiSquare.

Done

- Figure 1: Annotate pictures. Picture A would benefit from adding the names of the different containers. I'm not sure what the benefit of pictures B and C are (and not clear what I'm looking at in B).

Deleted Band C

- Figure 3A: Time should also be plotted on log-scale for consistency and ease of comparisons. Drawing means in different color or thicker lines helps for quick comparisons.

Done

Reviewer: 2

Comments to the Author(s)

This article presents an interesting research area in the use of monomolecular surface films for larviciding. However, the manuscript requires revisions. The introduction section of the manuscripts will require additional input to review in greater detail related published works. Also consider, discussing your results in much greater depth in the discussion section to allow interpretation of your research findings. Check on your references and ensure that appropriate referencing is done.

We have attempted to improve the detail in both the introduction and discussion section, we have also reread the cited references and ensured that they were appropriate (this required some deletions).

Appendix B

All reviewer comments are reproduced below in full in red

Thankyou to both reviewers for greatly improving this manuscript.

Reviewer: 1

Comments to the Author(s)

I would like to applaud the authors for incorporating the suggestions made on their manuscript. I believe the discussion is more useful now for the reader to interpret the data and the conclusions that are drawn are better incorporating the limitations of the experimental design.

I only have a few minor comments:

- The manuscript has greatly improved in its readability in terms of typos and grammatical mistakes, I would only suggest to focus on punctuation (eg comma placements).

We have had a native reader revise the manuscript including for grammar.

- I'm not too familiar with the type of analysis that has been done. However, looking at figures 2 and 3 and the 95% confidence intervals presented in the supplementary materials, I'm confused by the ultra-low p-values reported. For instance, there is quite some overlap between replicate 95CI, why these extremely low p-values? It would be useful to have the correctness of the statistical approach verified, particularly with these highly skewed log-scale data. However, I also note that these analyses will not change the conclusion.

We assume that the “the ultra-low p-values reported.” are those in table 1.. The values in table 1 are tests for whether the variables are explanatory across the entire dataset (both species). The 95% confidence intervals in figure 3 and the supplementary materials relate to the LT estimates for subsets of the data, consequently they are not directly comparable.

We have added the following text to the legend of table 1 to make the key distinction clearer for readers. Thanks for pointing this out.

Note that the P values in this table indicate whether the variable in question is likely to be an explanatory one, and do not directly relate to the confidence intervals for the LT estimates shown in figure 3.

Reviewer: 2

Comments to the Author(s)

There is a manuscript that provides important data on the use of surface films for mosquito control. The manuscript details the important contribution that surface films can play in the management of insecticide resistance, one of the major challenges in malaria control. Field studies would provide the required information on the operational use of the surface films for mosquito control.

Thankyou